# Ultra Fast Medoid Identification
# via Correlated Sequential Halving

**Tavor Z. Baharav**
Department of Electrical Engineering
Stanford University
Stanford, CA 94305
tavorb@stanford.edu

**David Tse**
Department of Electrical Engineering
Stanford University
Stanford, CA 94305
dntse@stanford.edu

## Abstract

The medoid of a set of $n$ points is the point in the set that minimizes the sum of distances to other points. It can be determined exactly in $O(n^2)$ time by computing the distances between all pairs of points. Previous works show that one can significantly reduce the number of distance computations needed by adaptively querying distances [1]. The resulting randomized algorithm is obtained by a direct conversion of the computation problem to a multi-armed bandit statistical inference problem. In this work, we show that we can better exploit the structure of the underlying computation problem by modifying the traditional bandit sampling strategy and using it in conjunction with a suitably chosen multi-armed bandit algorithm. Four to five orders of magnitude gains over exact computation are obtained on real data, in terms of both number of distance computations needed and wall clock time. Theoretical results are obtained to quantify such gains in terms of data parameters. Our code is publicly available online at `https://github.com/TavorB/Correlated-Sequential-Halving`.

## 1   Introduction

In large datasets, one often wants to find a single element that is representative of the dataset as a whole. While the mean, a point potentially outside the dataset, may suffice in some problems, it will be uninformative when the data is sparse in some domain; taking the mean of an image dataset will yield visually random noise [2]. In such instances the medoid is a more appropriate representative, where the medoid is defined as the point in a dataset which minimizes the sum of distances to other points. For one dimensional data under $\ell_1$ distance, this is equivalent to the median. This has seen use in algorithms such as $k$-medoid clustering due to its reduced sensitivity to outliers [3].

Formally, let $x_1, ..., x_n \in \mathcal{U}$, where the underlying space $\mathcal{U}$ is equipped with some distance function $d : \mathcal{U} \times \mathcal{U} \mapsto \mathbb{R}_+$. It is convenient to think of $\mathcal{U} = \mathbb{R}^d$ and $d(x, y) = \|x - y\|_2$ for concreteness, but other spaces and distance functions (which need not be symmetric or satisfy the triangle inequality) can be substituted. The medoid of $\{x_i\}_{i=1}^n$, assumed here to be unique, is defined as $x_{i^*}$ where

$$i^* = \operatorname*{argmin}_{i \in [n]} \theta_i \quad : \quad \theta_i \triangleq \frac{1}{n} \sum_{j=1}^n d(x_i, x_j) \tag{1}$$

Note that for non-adversarially constructed data, the medoid will almost certainly be unique. Unfortunately, brute force computation of the medoid becomes infeasible for large datasets, e.g. RNA-Seq datasets with $n = 100k$ points [4].

This issue has been addressed in recent works by noting that in most problem instances solving for the value of each $\theta_i$ exactly is unnecessary, as we are only interested in identifying $x_{i^*}$ and not in

computing every $\theta_i$ [1, 5, 6, 7]. This allows us to solve the problem by only estimating each $\theta_i$, such that we are able to distinguish with high probability whether it is the medoid. By turning this computational problem into a statistical one of estimating the $\theta_i$'s one can greatly decrease algorithmic complexity and running time. The key insight here is that sampling a random $J \sim \text{Unif}([n])$ and computing $d(x_i, x_J)$ gives an unbiased estimate of $\theta_i$. Clearly, as we sample and average over more independently selected $J_k \overset{iid}{\sim} \text{Unif}([n])$, we will obtain a better estimate of $\theta_i$. Estimating each $\theta_i$ to the same degree of precision by computing $\hat{\theta}_i = \frac{1}{T} \sum_{k=1}^T d(x_i, x_{J_k})$ yields an order of magnitude improvement over exact computation, via an algorithm like RAND [7].

In a recent work [1] it was observed that this statistical estimation could be done much more efficiently by adaptively allocating estimation budget to each of the $\theta_i$ in eq. (1). This is due to the observation that we only need to estimate each $\theta_i$ to a necessary degree of accuracy, such that we are able to say with high probability whether it is the medoid or not. By reducing to a stochastic multi-armed bandit problem, where each arm corresponds to a $\theta_i$, existing multi-armed bandit algorithms can be leveraged leading to the algorithm Med-dit [1]. As can be seen in Fig. 1 adding adaptivity to the statistical estimation problem yields another order of magnitude improvement.

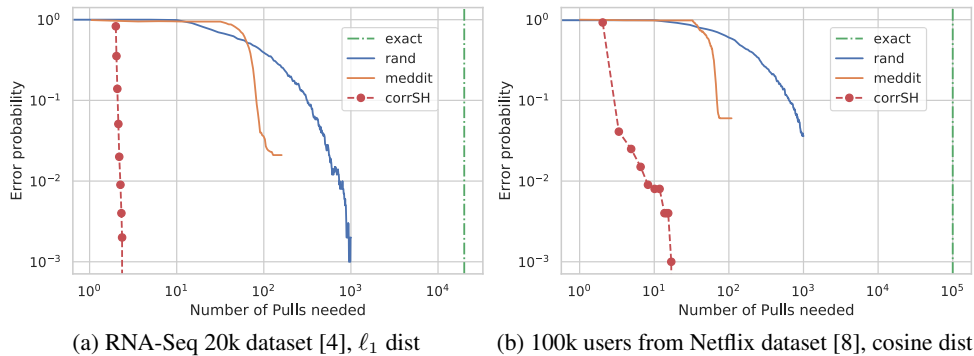

(a) RNA-Seq 20k dataset [4], $\ell_1$ dist      (b) 100k users from Netflix dataset [8], cosine dist

Figure 1: Empirical performance of exact computation, RAND, Med-dit and Correlated Sequential Halving The error probability is the probability of not returning the correct medoid.

## 1.1 Contribution

While adaptivity is already a drastic improvement, current schemes are still unable to process large datasets efficiently; running Med-dit on datasets with $n = 100k$ takes 1.5 hours. The main contribution of this paper is a novel algorithm that is able to perform this same computation in 1 minute. Our algorithm achieves this by observing that we want to find the minimum element and not the minimum value, and so our interest is only in the *relative* ordering of the $\theta_i$, not their actual values. In the simple case of trying to determine if $\theta_1 > \theta_2$, we are interested in estimating $\theta_1 - \theta_2$ rather than $\theta_1$ or $\theta_2$ separately. One can imagine the first step is to take one sample for each, i.e. $d(x_1, x_{J_1})$ to estimate $\theta_1$ and $d(x_2, x_{J_2})$ to estimate $\theta_2$, and compare the two estimates. In the direct bandit reduction used in the design of Med-dit, $J_1$ and $J_2$ would be *independently* chosen, since successive samples in the multi-armed bandit formulation are independent. In effect, we are trying to compare $\theta_1$ and $\theta_2$, but not using a common reference point to estimate them. This can be problematic for a sampling based algorithm, as it could be the case that $\theta_1 < \theta_2$, but the reference point $x_{J_1}$ we pick for estimating $\theta_1$ is on the periphery of the dataset as in Fig. 2a. This issue can fortunately be remedied by using the same reference point for both $x_1$ and $x_2$ as in Fig. 2b. By using the same reference point we are *correlating* the samples and intuitively reducing the variance of the estimator for $\theta_1 - \theta_2$. Here, we are exploiting the structure of the underlying computation problem rather than simply treating this as a standard multi-armed bandit statistical inference problem.

Building on this idea, we correlate the random sampling in our reduction to statistical estimation and design a new medoid algorithm, Correlated Sequential Halving. This algorithm is based on the Sequential Halving algorithm in the multi-armed bandit literature [9]. We see in Fig. 1 that we are able to gain another one to two orders of magnitude improvement, yielding an overall *four to five* orders of magnitude improvement over exact computation. This is accomplished by exploiting the fact that the underlying problem is computational rather than statistical.

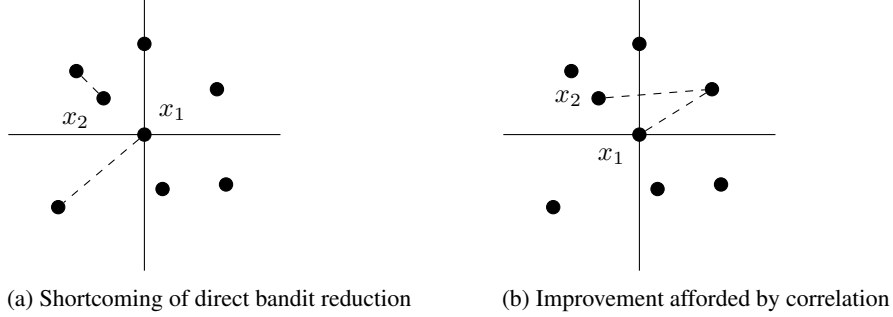

(a) Shortcoming of direct bandit reduction

(b) Improvement afforded by correlation

Figure 2: Toy 2D example

## 1.2 Theoretical Basis

We now provide high level insight into the theoretical basis for our observed improvement, later formalized in Theorem 2.1. We assume without loss of generality that the points are sorted so that $\theta_1 < \theta_2 \leq \ldots \leq \theta_n$, and define $\Delta_i \triangleq \theta_i - \theta_1$ for $i \in [n] \setminus \{1\}$, where $[n]$ is the set $\{1, 2, \ldots, n\}$. For visual clarity, we use the standard notation $a \vee b \triangleq \max(a, b)$ and $a \wedge b \triangleq \min(a, b)$, and assume a base of 2 for all logarithms .

Our proposed algorithm samples in a correlated manner as in Fig. 2b, and so we introduce new notation to quantify this improvement. As formalized later, $\rho_i$ is the improvement afforded by correlated sampling in distinguishing arm $i$ from arm 1. $\rho_i$ can be thought of as the relative reduction in variance, where a small $\rho_i$ indicates that $d(x_1, x_{J_1}) - d(x_i, x_{J_1})$ concentrates[1] faster than $d(x_1, x_{J_1}) - d(x_i, x_{J_2})$ about $-\Delta_i$ for $J_1, J_2$ drawn independently from $\mathrm{Unif}([n])$, shown graphically in Fig. 3.

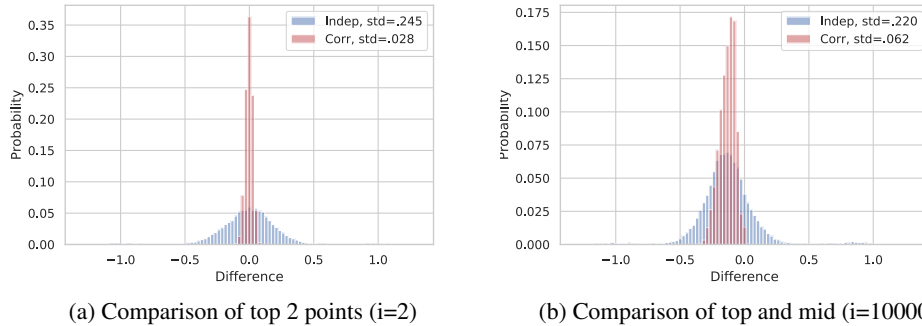

(a) Comparison of top 2 points (i=2)

(b) Comparison of top and mid (i=10000)

Figure 3: Correlated $d(1, J_1) - d(i, J_1)$ vs Independent $d(1, J_1) - d(i, J_2)$ sampling in RNA-Seq 20k dataset [4]. Averaged over the dataset, the independent samples have standard deviation $\sigma = 0.25$, so for (a) $\rho_i = .11$, and (b) $\rho_i = .25$

In the standard bandit setting with independent sampling, one needs a number of samples proportional to $H_2 = \max_{i \geq 2} i/\Delta_i^2$ to determine the best arm [10]. Replacing the standard arm difficulty of $1/\Delta_i^2$ with $\rho_i^2/\Delta_i^2$, the difficulty accounting for correlation, we show that one can solve the problem using a number of samples proportional to $\tilde{H}_2 = \max_{i \geq 2} i \rho_{(i)}^2 / \Delta_{(i)}^2$, an analogous measure. Here the permutation $(\cdot)$ indicates that the arms are sorted by decreasing $\rho_i/\Delta_i$ as opposed to just by $1/\Delta_i$. These details are formalized in Theorem 2.1.

Our theoretical improvement incorporating correlation can thus be quantified as $H_2/\tilde{H}_2$. As we show later in Fig. 5, in real datasets arms with small $\Delta_i$ have similarly small $\rho_i$, indicating that correlation yields a larger relative gain for previously difficult arms. Indeed, for the RNA-Seq 20k dataset we see that the ratio is $H_2/\tilde{H}_2 = 6.6$. The Netflix 100k dataset is too large to perform this calculation on, but for similar datasets like MNIST [11] this ratio is $4.8$. We hasten to note that this ratio does not

fully encapsulate the gains afforded by the correlation our algorithm uses, as only pairwise correlation is considered in our analysis. This is discussed further in Appendix B

## 1.3 Related Works

Several algorithms have been proposed for the problem of medoid identification. An $O(n^{3/2}2^{\Theta(d)})$ algorithm called TRIMED was developed finding the true medoid of a dataset under certain assumptions on the distribution of the points near the medoid [5]. This algorithm cleverly carves away non-medoid points, but unfortunately does not scale well with the dimensionality of the dataset. In the use cases we consider the data is very high dimensional, often with $d \approx n$. While this algorithm works well for small $d$, it becomes infeasible to run when $d > 20$. A similar problem, where the central vertex in a graph is desired, has also been analyzed. One proposed algorithm for this problem is RAND, which selects a random subset of vertices of size $k$ and measures the distance between each vertex in the graph and every vertex in the subset [7]. This was later improved upon with the advent of TOPRANK [6]. We build off of the algorithm Med-dit (*Med*oid-Ban*dit*), which finds the medoid in $\tilde{O}(n)$ time under mild distributional assumptions [1].

More generally, the use of bandits in computational problems has gained recent interest. In addition to medoid finding [1], other examples include Monte Carlo Tree Search for game playing AI [12], hyper-parameter tuning [13], $k$-nearest neighbor, hierarchical clustering and mutual information feature selection [14], approximate $k$-nearest neighbor [15], and Monte-Carlo multiple testing [16]. All of these works use a *direct* reduction of the computation problem to the multi-armed bandit statistical inference problem. In contrast, the present work further exploits the fact that the inference problem comes from a computational problem, which allows a more effective sampling strategy to be devised. This idea of preserving the structure of the computation problem in the reduction to a statistical estimation one has potentially broader impact and applicability to these other applications.

## 2 Correlated Sequential Halving

In previous works it was noted that sampling a random $J \sim \text{Unif}([n])$ and computing $d(x_i, x_J)$ gives an unbiased estimate of $\theta_i$ [1, 14]. This was where the problem was reduced to that of a multi-armed bandit and solved with an Upper Confidence Bound (UCB) based algorithm [17]. In their analysis, estimates of $\theta_i$ are generated as $\hat{\theta}_i = \frac{1}{|\mathcal{J}_i|} \sum_{j \in \mathcal{J}_i} d(x_i, x_j)$ for $\mathcal{J}_i \subseteq [n]$, and the analysis hinges on showing that as we sample the arms more, $\hat{\theta}_1 < \hat{\theta}_i \; \forall \, i \in [n]$ with high probability [2]. In a standard UCB analysis this is done by showing that each $\hat{\theta}_i$ individually concentrates. However on closer inspection, we see that this is not necessary; it is sufficient for the differences $\hat{\theta}_1 - \hat{\theta}_i$ to concentrate for all $i \in [n]$.

Using our intuition from Fig. 2 we see that one way to get this difference to concentrate faster is by sampling the same $j$ for both arms 1 and $i$. We can see that if $|\mathcal{J}_1| = |\mathcal{J}_i|$, one possible approach is to set $\mathcal{J}_1 = \mathcal{J}_i = \mathcal{J}$. This allows us to simplify $\hat{\theta}_1 - \hat{\theta}_i$ as

$$\hat{\theta}_1 - \hat{\theta}_i = \frac{1}{|\mathcal{J}_1|} \sum_{j \in \mathcal{J}_1} d(x_1, x_j) - \frac{1}{|\mathcal{J}_i|} \sum_{j \in \mathcal{J}_i} d(x_i, x_j) = \frac{1}{|\mathcal{J}|} \sum_{j \in \mathcal{J}} d(x_1, x_j) - d(x_i, x_j).$$

While UCB algorithms yield a serial process that samples one arm at a time, this observation suggests that a different algorithm that pulls many arms at the same time would perform better, as then the same reference $j$ could be used. By estimating each points' centrality $\theta_i$ independently, we are ignoring the dependence of our estimators on the random reference points selected; using the same set of reference points for estimating each $\theta_i$ reduces the variance in the choice of random reference points. We show that a modified version of *Sequential Halving* [10] is much more amenable to this type of analysis. At a high level this is due to the fact that Sequential Halving proceeds in stages by sampling arms uniformly, eliminating the worse half of arms from consideration, and repeating. This very naturally obeys this "correlated sampling" condition, as we can now use the same set of reference points $\mathcal{J}$ for all arms under consideration in each round. We present the slightly modified

algorithm below, introducing correlation and capping the number of pulls per round, noting that the main difference comes in the analysis rather than the algorithm itself.

---

**Algorithm 1** Correlated Sequential Halving

---

1: **Input:** Sampling budget $T$, dataset $\{x_i\}_{i=1}^n$
2: initialize $S_0 \leftarrow [n]$
3: **for** r=0 **to** $\lceil \log n \rceil - 1$ **do**
4:     select a set $\mathcal{J}_r$ of $t_r$ data point indices uniformly
       at random without replacement from $[n]$ where

$$t_r = \left\{ 1 \vee \left\lfloor \frac{T}{|S_r| \lceil \log n \rceil} \right\rfloor \right\} \wedge n$$

5:     For each $i \in S_r$ set $\hat{\theta}_i^{(r)} = \frac{1}{t_r} \sum_{j \in \mathcal{J}_r} d(x_i, x_j)$
6:     **if** $t_r = n$ **then**
7:        Output arm in $S_r$ with the smallest $\hat{\theta}_i^{(r)}$
8:     **else**
9:        Let $S_{r+1}$ be the set of $\lceil |S_r|/2 \rceil$ arms in $S_r$ with the smallest $\hat{\theta}_i^{(r)}$
10:    **end if**
11: **end for**
12: **return** arm in $S_{\lceil \log n \rceil}$

---

Examining the random variables $\hat{\Delta}_i \triangleq d(x_1, x_J) - d(x_i, x_J)$ for $J \sim \text{Unif}([n])$, we see that for any fixed dataset all $\hat{\Delta}_i$ are bounded, as $\max_{i,j \in [n]} d(x_i, x_j)$ is finite. In particular, this means that all $\hat{\Delta}_i$ are sub-Gaussian.

**Definition 1.** *We define $\sigma$ to be the minimum sub-Gaussian constant of $d(x_I, x_J)$ for $I, J$ drawn independently from Unif([n]). Additionally, for $i \in [n]$ we define $\rho_i \sigma$ to be the minimum sub-Gaussian constant of $d(x_1, x_J) - d(x_i, x_J)$, where $\sigma$ is as above and $\rho_i$ is an arm (point) dependent scaling, as displayed in Figure 3.*

This shifts the direction of the analysis, as where in previous works the sub-Gaussianity of $d(x_1, x_J)$ was used [1], we now instead utilize the sub-Gaussianity of $d(x_1, x_J) - d(x_i, x_J)$. Here $\rho_i \leq 1$ indicates that the correlated sampling improves the concentration and by extension the algorithmic performance.

A standard UCB algorithm is unable to algorithmically make use of these $\{\rho_i\}$. Even considering batch UCB algorithms, in order to incorporate correlation the confidence bounds would need to be calculated differently for each pair of arms depending on the number of $j$'s they've pulled in common and the sub-Gaussian parameter of $d(x_{i_1}, x_J) - d(x_{i_2}, x_J)$. It is unreasonable to assume this is known for all pairs of points a priori, and so we restrict ourselves to an algorithm that only uses these pairwise correlations implicitly in its analysis instead of explicitly in the algorithm. Below we state the main theorem of the paper.

**Theorem 2.1.** *Assuming that $T \geq n \log n$ and denoting the sub-Gaussian constants of $d(x_1, x_J) - d(x_i, x_J)$ as $\rho_i \sigma$ for $i \in [n]$ as in definition 1, Correlated Sequential Halving (Algorithm 1) correctly identifies the medoid in at most $T$ distance computations with probability at least*

$$1 - 3 \log n \exp \left( -\frac{T}{16\sigma^2 \log n} \cdot \min_{i \geq \frac{T}{n \log n}} \left[ \frac{\Delta_{(i)}^2}{i \rho_{(i)}^2} \right] \right)$$

*which can be coarsely lower bounded as*  $\qquad 1 - 3 \log n \cdot \exp \left( -\frac{T}{16 \tilde{H}_2 \sigma^2 \log n} \right)$

*where* $\qquad \tilde{H}_2 = \max_{i \geq 2} \frac{i \rho_{(i)}^2}{\Delta_{(i)}^2} \qquad , \qquad (\cdot) : [n] \mapsto [n], (1) = 1, \frac{\Delta_{(2)}}{\rho_{(2)}} \leq \frac{\Delta_{(3)}}{\rho_{(3)}} \leq \cdots \leq \frac{\Delta_{(n)}}{\rho_{(n)}}$

Above $\tilde{H}_2$ is a natural measure of hardness for this problem analogous to $H_2 = \max_i \frac{i}{\Delta_i^2}$ in the standard bandit case, and $(\cdot)$ orders the arms by difficulty in distinguishing from the best arm after taking into account the $\rho_i$. We defer the proof of Thm. 2.1 and necessary lemmas to Appendix A for readability.

## 2.1 Lower bounds

Ideally in such a bandit problem, we would like to provide a matching lower bound. We can naively lower bound the sample complexity as $\Omega(n)$, but unfortunately no tighter results are known. A more traditional bandit lower bound was recently proved for adaptive sampling in the approximate $k$-NN case, but requires that the algorithm only interact with the data by sampling coordinates uniformly at random [15]. This lower bound can be transferred to the medoid setting, however this constraint becomes that an algorithm can only interact with the data by measuring the distance from a desired point to another point selected uniformly at random. This unfortunately removes all the correlation effects we analyze. For a more in depth discussion of the difficulty of providing a lower bound for this problem and the higher order problem structure causing this, we refer the reader to Appendix B.

## 3 Simulation Results

Correlated Sequential Halving (corrSH) empirically performs much better than UCB type algorithms on all datasets tested, reducing the number of comparisons needed by 2 orders of magnitude for the RNA-Seq dataset and by 1 order of magnitude for the Netflix dataset to achieve comparable error probabilities, as shown in Table 1. This yields a similarly drastic reduction in wall clock time which contrasts most UCB based algorithms; usually, when implemented, the overhead needed to run UCB makes it so that even though there is a significant reduction in number of pulls, the wall clock time improvement is marginal [14].

| dataset, metric | $n$, $d$ | | corrSH | Med-dit | Rand | Exact Comp. |
|---|---|---|---|---|---|---|
| RNA-Seq 20k, $\ell_1$ | 20k, 28k | time | **10.9** | 246 | 2131 | 40574 |
| | | # pulls | **2.43** | 121 (2.1%) | 1000 (.1%) | 20000 |
| RNA-Seq 100k, $\ell_1$ | 109k, 28k | time | **64.2** | 5819 | 10462 | - |
| | | # pulls | **2.10** | 420 | 1000 (.5%) | 100000 |
| Netflix 20k, cosine dist | 20k, 18k | time | **6.82** | 593 | 70.2 | 139 |
| | | # pulls | **15.0** | 85.8 | 1000 (.6%) | 20000 |
| Netflix 100k, cosine dist | 100k, 18k | time | **53.4** | 6495 | 959 | - |
| | | # pulls | **18.5** | 90.5 (6%) | 1000 (3.6%) | 100000 |
| MNIST Zeros, $\ell_2$ | 6424, 784 | time | **1.46** | 151 | 65.7 | 22.8 |
| | | # pulls | **47.9** | 91.2 (.1%) | 1000 (65.2%) | 6424 |

Table 1: Performance in average number of pulls per arm. Final percent error noted parenthetically if nonzero. corrSH was run with varying budgets until it had no failures on the 1000 trials.

We note that in our simulations we only used 1 pull to initialize each arm for Med-dit for plotting purposes where in reality one would use 16 or some larger constant, sacrificing a small additional number of pulls for a roughly $10\%$ reduction in wall clock time. In these plots we show a comparison between Med-dit [1], Correlated Sequential Halving, and RAND [7], shown in Figures 1 and 4.

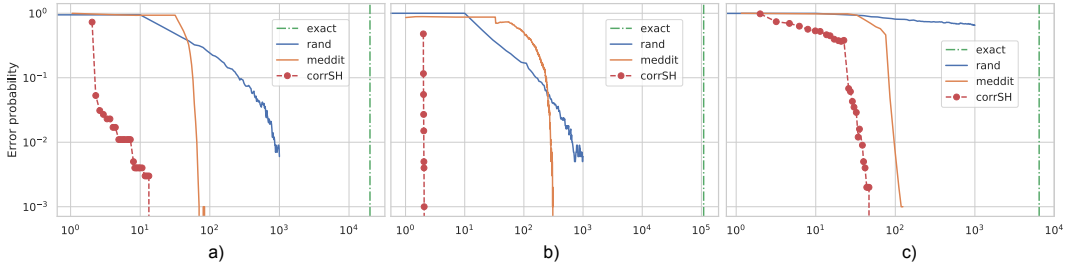

Figure 4: Number of pulls versus error probability for various datasets and distance metrics. (a) Netflix 20k, cosine [8]. (b) RNA-Seq 100k, $\ell_1$ [4] (c) MNIST, $\ell_2$ [11]

## 3.1 Simulation details

The 3 curves for the randomized algorithms previously discussed are generated in different ways. For RAND and Med-dit the curves represent the empirical probability, averaged over 1000 trials, that after $nx$ pulls ($x$ pulls per arm on average) the true medoid was the empirically best arm. RAND was run with a budget of 1000 pulls per arm, and Med-dit was run with target error probability of $\delta = 1/n$. Since Correlated Sequential Halving behaves differently after $x$ pulls per arm depending on what its input budget was, it requires a different method of simulation; every solid dot in the plots represents the average of 1000 trials at a fixed budget, and the dotted line connecting them is simply interpolating the projected performance. In all cases the only variable across trials was the random seed, which was varied across 0-999 for reproducibility. The value noted for Correlated Sequential Halving in Table 1 is the minimum budget above which all simulated error probabilities were 0.

**Remark 1.** *In theory it is much cleaner to discard samples from previous stages when constructing the estimators in stage $r$ to avoid dependence issues in the analysis. In practice we use these past samples, that is we construct our estimator for arm $i$ in stage $r$ from all the samples seen of arm $i$ so far, rather than just the $t_r$ fresh ones.*

Many different datasets and distance metrics were used to validate the performance of our algorithm. The first dataset used was a single cell RNA-Seq one, which contains the gene expressions corresponding to each cell in a tissue sample. A common first step in analyzing single cell RNA-Seq datasets is clustering the data to discover sub classes of cells, where medoid finding is used as a subroutine. Since millions of cells are sequenced and tens of thousands of gene expressions are measured in such a process, this naturally gives us a large high dimensional dataset. Since the gene expressions are normalized to a probability distribution for each cell, $\ell_1$ distance is commonly used for clustering [18]. We use the 10xGenomics dataset consisting of 27,998 gene-expressions over 1.3 million neuron cells from the cortex, hippocampus, and subventricular zone of a mouse brain [4]. We test on two subsets of this dataset, a small one of 20,000 cells randomly subsampled, and a larger one of 109,140 cells, the largest true cluster in the dataset. While we can exactly compute a solution for the 20k dataset, it is computationally difficult to do so for the larger one, so we use the most commonly returned point from our algorithms as ground truth (all 3 algorithms have the same most frequently returned point).

Another dataset we used was the famous Netflix-prize dataset [8]. In such recommendation systems, the objective is to cluster users with similar preferences. One challenge in such problems is that the data is very sparse, with only .21% of the entries in the Netflix-prize dataset being nonzero. This necessitates the use of normalized distance measures in clustering the dataset, like cosine distance, as discussed in [2, Chapter 9]. This dataset consists of 17,769 movies and their ratings by 480,000 Netflix users. We again subsample this dataset, generating a small and large dataset of 20,000 and 100,000 users randomly subsampled. Ground truth is generated as before.

The final dataset we used was the zeros from the commonly used MNIST dataset [11]. This dataset consists of centered images of handwritten digits. We subsample this, using only the images corresponding to handwritten zeros, in order to truly have one cluster. We use $\ell_2$ distance, as root mean squared error (RMSE) is a frequently used metric for image reconstruction. Combining the train and test datasets we get 6,424 images, and since each image is 28x28 pixels we get $d = 784$. Since this is a smaller dataset, we are able to compute the ground truth exactly.

## 3.2 Discussion on $\{\rho_i\}$

For correlation to improve our algorithmic performance, we ideally want $\rho_i \ll 1$ and decaying with $\Delta_i$. Empirically this appears to be the case as seen in Fig. 5, where we plot $\rho_i$ versus $\Delta_i$ for the RNA-Seq and MNIST datasets. $\frac{1}{\rho_i^2}$ can be thought of as the multiplicative reduction in number of pulls needed to differentiate that arm from the best arm, i.e. $\frac{1}{\rho_i} = 10$ roughly implies that we need a factor of 100 fewer pulls to differentiate it from the best arm due to our "correlation". Notably, for arms that would normally require many pulls to differentiate from the best arm (small $\Delta_i$), $\rho_i$ is also small. Since algorithms spend the bulk of their time differentiating between the top few arms, this translates into large practical gains.

One candidate explanation for the phenomena that small $\Delta_i$ lead to small $\rho_i$ is that the points themselves are close in space. However, this intuition fails for high dimensional datasets as shown in

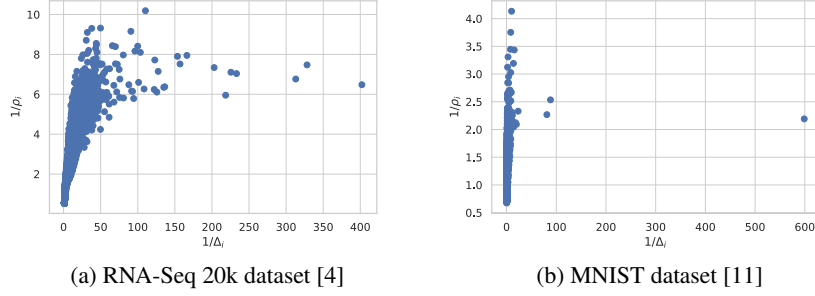

(a) RNA-Seq 20k dataset [4]

(b) MNIST dataset [11]

Figure 5: $1/\Delta_i$ vs. $1/\rho_i$ in real world dataset

Fig. 6. We do see empirically however that $\rho_i$ decreases with $\Delta_i$, which drastically decreases the number of comparisons needed as desired.

We can bound $\rho_i$ if our distance function obeys the triangle inequality, as $\hat{\Delta}_i \triangleq d(x_i, x_J) - d(x_1, x_J)$ is then a bounded random variable since $|\hat{\Delta}_i| \le d(x_i, x_1)$. Combining this with the knowledge that $\mathbb{E}\hat{\Delta}_i = \Delta_i$ we get $\hat{\Delta}_i$ is sub-Gaussian with parameter at most

$$\rho_i \sigma \le \frac{2d(x_i, x_1) + \Delta_i}{2}$$

Alternatively, if we assume that $\hat{\Delta}_i$ is normally distributed with variance $\rho_i^2 \sigma^2$, we are able to get a tighter characterization of $\rho_i$:

$$\begin{aligned}
\rho_i^2 \sigma^2 &= \mathrm{Var}(d(1, J) - d(i, J)) \\
&= \mathbb{E}\left[ (d(1, J) - d(i, J))^2 \right] - \left( \mathbb{E}\left[ d(1, J) - d(i, J) \right] \right)^2 \\
&\le d(1, i)^2 - \Delta_i^2
\end{aligned}$$

We can clearly see that as $d(1, i) \to 0$, $\rho_i$ decreases, to 0 in the normal case. However in high dimensional datasets $d(1, i)$ is usually not small for almost any $i$. This is empirically shown in Fig. 6.

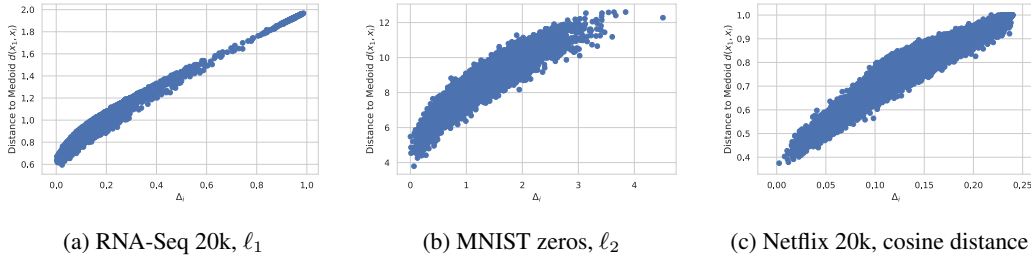

(a) RNA-Seq 20k, $\ell_1$

(b) MNIST zeros, $\ell_2$

(c) Netflix 20k, cosine distance

Figure 6: Distance from point $i$ to the medoid, $d(x_1, x_i)$ versus $\Delta_i$

While $\rho_i$ can be small, it is not immediately clear that it is bounded above. However, since our distances are bounded for any given dataset, we have that $d(1, J)$ and $d(i, J)$ are both $\sigma$-sub-Gaussian for some $\sigma$, and so we can bound the sub-Gaussian parameter of $d(1, J) - d(i, J)$ quantity using the Orlicz norm.

$$\rho_i^2 \sigma^2 = \|d(1, J) - d(i, J)\|_\Psi^2 \le (\|d(1, J)\|_\Psi + \|d(i, J)\|_\Psi)^2 = 4\sigma^2$$

While this appears to be worse at first glance, we are able to jointly bound $\mathbb{P}(\hat{\theta}_i - \hat{\theta}_1 - \Delta_i < -\Delta_i) \le \exp\left( -\frac{n\Delta_i^2}{2\rho_i^2 \sigma^2} \right) \le \exp\left( -\frac{n\Delta_i^2}{8\sigma^2} \right)$ by the control of $\rho_i$ above. In the regular case, this bound is achieved by separating the two and bounding the probability that either $\hat{\theta}_i < \theta_i - \Delta_i/2$ or $\hat{\theta}_1 > \theta_1 + \Delta_i/2$, which yields an equivalent probability since we need $\hat{\theta}_i, \hat{\theta}_1$ to concentrate to half the original width. Hence, even for data without significant correlation, attempting to correlate the noise will not increase the number of pulls required when using this standard analysis method.

### 3.3 Fixed Budget

In simulating Correlated Sequential Halving, we swept the budget over a range and reported the smallest budget above which there were 0 errors in 1000 trials. One logical question given a fixed budget algorithm like corrSH is then, for a given problem, what to set the budget to. This is an important question for further investigation, as there does not seem to be an efficient way to estimate $\tilde{H}_2$. Before providing our doubling trick based solution, we would like to note that it is unclear what to set the hyperparameters to for any of the aforementioned randomized algorithms. RAND is similarly fixed budget, and for Med-dit, while setting $\delta = \frac{1}{n}$ achieves vanishing error probability theoretically, using this setting in practice for finite $n$ yields an error rate of $6\%$ for the Netflix 100k dataset. Additionally, the fixed budget setting makes sense in the case of limited computed power or time sensitive applications.

The approach we propose is a variant of the doubling trick, which is commonly used to convert fixed budget or finite horizon algorithms to data dependent or anytime ones. Here this would translate to running the algorithm with a certain budget $T$ (say $3n$), then doubling the budget to $6n$ and rerunning the algorithm. If the two answers are the same, declare this the medoid and output it. If the answers are different, double the budget again to $12n$ and compare. The odds that the same incorrect arm is outputted both times is exceedingly small, as even with a budget that is too small, the most likely output of this algorithm is the true medoid. This requires a budget of at most $8T$ to yield approximately the same error probability as that of just running our algorithm with budget $T$.

## 4  Summary

We have presented a new algorithm, Correlated Sequential Halving, for computing the medoid of a large dataset. We prove bounds on it's performance, deviating from standard multi-armed bandit analysis due to the correlation in the arms. We include experimental results to corroborate our theoretical gains, showing the massive improvement to be gained from utilizing correlation in real world datasets. There remains future practical work to be done in seeing if other computational or statistical problems can benefit from this correlation trick. Additionally there are open theoretical questions in proving lower bounds for this special query model, seeing if there is any larger view of correlation beyond pairwise that is analytically tractable, and analyzing this generalized stochastic bandits setting.

## Acknowledgements

The authors gratefully acknowledge funding from the NSF GRFP, Alcatel-Lucent Stanford Graduate Fellowship, NSF grant under CCF-1563098, and the Center for Science of Information (CSoI), an NSF Science and Technology Center under grant agreement CCF-0939370.

## Footnotes

[1]Throughout this work we talk about concentration in the sense of the empirical average of a random variable concentrating about the true mean of that random variable.

[2]In order to maintain the unbiasedness of the estimator given the sequential nature of UCB, reference points are chosen with replacement in Med-dit, potentially yielding a multiset $\mathcal{J}_i$. For the sake of clarity we ignore this subtlety for Med-dit, as our algorithm samples without replacement.

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
