[Supplementary Material]

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

# Appendices

## A  Proof of Theorem 2.1

We assume $n$ is a power of 2 for readability, but the analysis holds for any $n$. We begin with the following immediate consequence of Hoeffding's inequality, remembering that $|\mathcal{J}_r| = t_r$:

**Lemma A.1.** *Assume that the best arm was not eliminated prior to round r. Then for any arm $i \in S_r$*

$$\mathbb{P}\left(\hat{\theta}_1^{(r)} > \hat{\theta}_i^{(r)}\right) = \mathbb{P}\left(\frac{1}{|\mathcal{J}_r|}\sum_{j \in \mathcal{J}_r} d(x_1, x_j) - d(x_i, x_j) + \Delta_i > \Delta_i\right) \le \exp\left(\frac{-t_r \Delta_i^2}{2\rho_i^2 \sigma^2}\right)\mathbb{I}\{t_r < n\}$$

*Where if $t_r = n$ we know that this probability is exactly 0 by definition of the medoid.*

We now examine one round of Correlated Sequential Halving and bound the probability that the algorithm eliminates the best arm at round $r$, recalling that

$$t_r = \left\{1 \vee \left\lfloor \frac{T}{|S_r|\lceil \log n \rceil}\right\rfloor\right\} \wedge n.$$

**Lemma A.2.** *The probability that the medoid is eliminated in round $r$ is at most*

$$3\exp\left(-\frac{T}{8\sigma^2 \log n} \cdot \left\lceil \frac{\Delta_{(i_r)}^2}{i_r \rho_{(i_r)}^2}\right\rceil\right)\mathbb{I}\{t_r < n\}$$

*for $i_r = |S_r|/4 = \frac{n}{2^{r+2}}$*

*Proof.* The proof follows similarly to that of [10], modulo the interesting feature that if $t_r = n$ there is no uncertainty. Additionally, the analysis differs in that here we are interested in giving the sample complexity in terms of $\frac{\Delta_{(i)}}{\rho_{(i)}}$ instead of $\Delta_i$, and so instead of removing arms $i$ with low $\Delta_i$ from consideration as in [10], we remove arms with low $\frac{\Delta_i}{\rho_i}$ for the analysis.

Formally, define $S_r'$ as the set of arms in $S_r$ excluding the $i_r = \frac{1}{4}|S_r|$ arms $i$ with smallest $\frac{\Delta_i}{\rho_i}$.

We define the random variable $N_r$ as the number of arms in $S_r'$ whose empirical average in round $r$, $\hat{\theta}_i^{(r)}$, is smaller than that of the optimal arm. We begin by showing that $\mathbb{E}[N_r]$ is small.

$$\mathbb{E}[N_r] = \sum_{j \in S_r'} \mathbb{P}\left(\hat{\theta}_i^{(r)} > \hat{\theta}_j^{(r)}\right)$$

$$\le \sum_{j \in S_r'} \exp\left(-\frac{t_r \Delta_j^2}{2\rho_j^2 \sigma^2}\right)\mathbb{I}\{t_r < n\}$$

$$\le \sum_{j \in S_r'} \exp\left(-\frac{T\Delta_j^2}{4\rho_j^2 \sigma^2 |S_r| \log n}\right)\mathbb{I}\{t_r < n\}$$

$$\le |S_r'| \max_{j \in S_r'} \exp\left(-\frac{T\Delta_j^2}{4\rho_j^2 \sigma^2 |S_r| \log n}\right)\mathbb{I}\{t_r < n\}$$

$$= |S_r'| \exp\left(-\frac{T}{16\sigma^2 \log n} \cdot \frac{1}{i_r} \cdot \min_{j \in S_r'}\left\{\frac{\Delta_j^2}{\rho_j^2}\right\}\right)\mathbb{I}\{t_r < n\}$$

$$\le |S_r'| \exp\left(-\frac{T}{16\sigma^2 \log n} \cdot \frac{1}{i_r} \cdot \min_{i \ge i_r}\left\{\frac{\Delta_{(i)}^2}{\rho_{(i)}^2}\right\}\right)\mathbb{I}\{t_r < n\}$$

$$= |S_r'| \exp\left(-\frac{T}{16\sigma^2 \log n} \cdot \left\lceil\frac{\Delta_{(i_r)}^2}{i_r \rho_{(i_r)}^2}\right\rceil\right)\mathbb{I}\{t_r < n\}$$

Where in the third line we assumed $T \geq n \log n$ so that

$$t_r = \left\lfloor \frac{T}{|S_r|\lceil \log n \rceil} \right\rfloor \geq \frac{T}{2|S_r|\lceil \log n \rceil}$$

Additionally, in the second to last line we used the fact that due to the removal of arms with small $\frac{\Delta_{(i)}}{\rho_{(i)}}$, for all arms $j \in S'_r$ where $j = (i)$, we have that $i \geq i_r$.

We now see that in order for the best arm to be eliminated in round $r$ at least $|S_r|/2$ arms must have lower empirical averages in round $r$. This means that at least $|S_r|/4$ arms from $S'_r$ must outperform the best arm, i.e. $N_r \geq |S_r|/4 = |S'_r|/3$.

We can then bound this probability with Markov's inequality as below:

$$\mathbb{P}\left( N_r \geq \frac{1}{3}|S'_r| \right) \leq 3\mathbb{E}[N_r]/|S'_r| \leq 3\exp\left( -\frac{T}{16\sigma^2 \log n} \cdot \left[ \frac{\Delta^2_{(i_r)}}{i_r \rho^2_{(i_r)}} \right] \right)\mathbb{I}\{t_r < n\}.$$

$\square$

We note that $t_r < n$ is a deterministic condition. Via some algebra, we obtain that

$$t_r = \left\lfloor \frac{T}{|S_r|\lceil \log n \rceil} \right\rfloor \leq \frac{T}{|S_r| \log n} = \frac{T2^r}{n \log n}$$

This means that if $r < \log\left( \frac{n^2 \log n}{T} \right)$ then $t_r < n$. To this end we define $r_{max} \triangleq \left\lfloor \log\left( \frac{n^2 \log n}{T} \right) \right\rfloor$ and $i_{r_{max}} \triangleq \frac{n}{2^{r_{max}}} \geq \frac{T}{n \log n}$. With this in place, we are now able to easily prove Theorem 2.1

*Proof.* The algorithm clearly does not exceed its budget of $T$ arm pulls (distance measurements). Further, if the best arm survives the execution of all $\log n$ rounds then the algorithm succeeds as all other arms must have been eliminated. Hence, by a union bound over the stages, our probability of failure (the best arm being eliminated in any of the $\log n$ stages) is at most

$$3\sum_{r=1}^{\log n} \exp\left( -\frac{T}{16\sigma^2 \log n} \cdot \left[ \frac{\Delta^2_{(i_r)}}{i_r \rho^2_{(i_r)}} \right] \right)\mathbb{I}\{t_r < n\}$$

$$\leq 3\sum_{r=1}^{\log n} \exp\left( -\frac{T}{16\sigma^2 \log n} \cdot \left[ \frac{\Delta^2_{(i_r)}}{i_r \rho^2_{(i_r)}} \right] \right)\mathbb{I}\left\{ r < \log\left( \frac{n^2 \log n}{T} \right) \right\}$$

$$\leq 3\log n \exp\left( -\frac{T}{16\sigma^2 \log n} \cdot \min_{r \leq r_{max}} \left[ \frac{\Delta^2_{(i_r)}}{i_r \rho^2_{(i_r)}} \right] \right)$$

$$\leq 3\log n \exp\left( -\frac{T}{16\sigma^2 \log n} \cdot \min_{i \geq i_{r_{max}}} \left[ \frac{\Delta^2_{(i)}}{i \rho^2_{(i)}} \right] \right)$$

$$\leq 3\log n \cdot \exp\left( -\frac{T}{16\tilde{H}_2 \sigma^2 \log n} \right)$$

We note that in cases where $\frac{\rho^2_{(i)}}{\Delta^2_{(i)}}$ is very large for small $i$, this last line is loose.

$\square$

**Remark 2.** *A standard analysis of this algorithm, ignoring arms with small $\Delta_i$ to create $S'_r$, would yield hardness measure $H'_2 = \max_i \frac{i \rho^2_i}{\Delta^2_i}$. However, we can see by pigeonhole principle that*

$$\max_i \frac{i \rho^2_i}{\Delta^2_i} \geq \max_i \frac{i \rho^2_{(i)}}{\Delta^2_{(i)}}$$

**Remark 3.** *While it is convenient to think of $\mathcal{U} = \mathbb{R}^d$ and $d(x, y) = \|x - y\|_2$, we note that our results are valid for arbitrary distance functions which may not be symmetric or obey the triangle inequality, like Bregman divergences or squared Euclidean distance.*

# B Lower bounds

It seems very difficult to generate lower bounds for the sample complexity of the medoid problem due to the higher order structure present.

## B.1 Beyond pairwise correlation

Throughout this work we have discussed the benefits of correlating measurements. However, the only way in which correlation figured into our analysis was in helping $\hat{\theta}_i - \hat{\theta}_1$ concentrate. Due to this correlation we can show that the difference between estimators concentrates quickly, analyzing pairs of estimators rather than just individual $\hat{\theta}_i$. This leads to the natural question, can correlation help beyond just pairs of estimators?

We answer this question in the affirmative. As a concrete example assume that $\{x_i\}_{i=1}^n \in \mathbb{R}^2$ are evenly spaced around the unit circle, and $x_0 = (0, 0)$ is the medoid of $\{x_i\}_{i=0}^n$. For a reference point $x_J$ drawn uniformly at random we define $\hat{\Delta}_i \triangleq d(x_i, x_J) - d(x_1, x_J)$.

Let $x_i = (1, 0)$, $x_k = (-1, 0)$. We have previously shown that $\hat{\Delta}_i, \hat{\Delta}_k$ concentrate nicely. However, in sequential halving, we are concerned with the probability that over half the estimators appear better than the best estimator, i.e. $\hat{\Delta}_i < 0$ for many $i$ (more than $n/2$ for the first round). Many samples are needed to argue that this is small if we assume that the events $\hat{\Delta}_i < 0$ and $\hat{\Delta}_k < 0$ are independent as is currently being done, but we can clearly see that for $i, k$ as given, $\mathbb{P}\left(\{\hat{\Delta}_i < 0\} \cap \{\hat{\Delta}_k < 0\}\right) = 0$ where the probability is taken with respect to the randomness in selecting a common reference point $x_J$.

It is not clear what quantities we should be interested in when looking at all the estimators jointly, but it is clear that there are additional benefits to correlation beyond simply the improved concentration of differences of estimators.

## B.2 Bandit lower bounds

Ideally in such a bandit problem we would like to provide a matching lower bound. This is made difficult by the fact that we lack insight into which quantities are relevant in determining the hardness of the problem. A more traditional bandit lower bound was recently proved for adaptive sampling in the approximate $k$-NN case, but this lower bound requires the data points to be constrained, i.e. $[x_i]_j \in \{\pm 1/2\}$, and more importantly that the algorithm is only allowed to interact with the data by sampling coordinates uniformly at random [15]. This second constraint on the algorithm unfortunately removes all the structure we wish to analyze from the problem. The lower bound is

proved using a change of measure argument, neatly presented in [9]. In the case we wish to analyze, strategies are no longer limited to random sampling the data, i.e. for a given $x_i$ we can measure its distance to a specific $x_j$, we do not need to independently sample a reference point for each pull.

Currently, we do not know of any data dependent lower bound for this problem. A trivial lower bound is $\Omega(n)$ distance computations, as we need to perform at least one distance computation for every data point. However, we have as of yet been unable to provide any tighter lower bounds in terms of the $\rho_i$'s or any larger scale structure as mentioned above.