[Reviews · NeurIPS 2019]

Reviewer 1



The introduction and general idea behind the paper is written in a way that is easy to understand and logically structured. There are, though, a few places were clarity could be improved. Most of these are around the plot in Fig. 4 which is often referenced as showcasing particular properties, especially in 3.2. This is in part due to the content and meaning of the figure never being fully and completely being introduced, doing so would be beneficial for a figure which appears so central to the paper. Another aspect that would have been nice to see is a comparison of the theoretical results with the practical ones, i.e. it would be interesting to see how loose the bound from Theorem 2.1 is in practice, at least for the datasets used. In the experiments it is mentioned that 1000 runs were performed for every experiment, yet only the average value is reported. While the log scale might make the variance of those values hard to impossible to visualize providing some intuition on the variability in the results would be appreciated.

Reviewer 2



This paper proposes a randomized method that reduces the number of distance computations needed to identify the medoid of a set. Earlier results are based on reducing the computational problem into a statistical inference problem (multi-armed bandit). This paper is farther extending this technique by exploits the underlying structure of the computational problem. This is done by correlating the samples, i.e., using the same samples to estimate a set of estimators. The authors claim that in many real-world problems this technique is reducing the variance of the estimators which leads to better results. Indeed, the experimental results show a big improvement in comparison with other existing algorithms. In addition, the authors show data dependent theoretical result under the assumption that the estimators are sub-Gaussian. -The paper is really well-written and clearly structured. I have really enjoyed reading it. Especially the introduction and related work sections. They are doing great work describing the novelty, originality, and main ideas of the paper. Moreover, I liked the figures and their locations inside the paper. Well done! -I briefly read the proof of the main theorem (It is inside the supplementary). It appears to be correct. I think that finding assumptions on the distribution of the data or the metric space, which guarantee a small number of distance computations (small H_2) can really strengthen the theoretic part. Moreover, I believe the assumptions of the theorem should be stated clearly inside the theorem (T>=nlogn and the main assumption). -In my opinion, the notion and results introduced in the paper are important. According to the experiments, it seems that the suggested algorithm obtains a big improvement on some interesting data sets. Moreover, I believe that due to the good presentation and the simple ideas of this paper, practitioners and researchers can use this idea in other domains. Edit: I have read the author response. It seems the author understood my comments and is planning to fix what's necessary for the final version. Other comments: - line 115: The footnote notation looks like the notation of the square sign (^2). It confused me and it took me a while to understand this. - typo: equation under line 122: x_2->x_i -Inside the algorithm: T is not defined. Should be written as an input to the algorithm. It took me time to understand that this is the budget. - Table 1: What is the error rate of corrSH? does it missing? - Line 249: What is the middle of the road point? - in my opinion, remark 3 should be inside the main paper. Observation and discussion are important to the reader.

Reviewer 3



This submission attempts to improve an existing randomized algorithm for estimating the medoid of a set of points using a multi-armed bandit technique by taking advantage of sampling distances using the same reference point. Intuitively, this improved technique makes sense. They show that the assumption that looking at the difference of their random variables improves concentration (Fig 4). Through both theoretical guarantees (The 2.1) and experiments they show the benefits of this sampling technique paired with a suitable multi-armed bandit algorithm. While the idea of correlated sampling is obviously nothing new, this technique has not yet been applied to the medoid problem. The mathematical content is of a relatively high quality. The precision of writing leaves something to be desired. In a number of places there are grammatical errors and the phrasing is somewhat sloppy. This is a minor detail, but giving the manuscript to a proofreader proficient in technical writing would benefit this work. This algorithm is one that is likely to be used in practice for estimating the medoid of a dataset when the cost of an exact solution (O(n^2)) is too expensive. However, I am not fully qualified to judge how often such a situation arises, or the importance of the problem of computing the medoid of a dataset. Including references or specific examples in which this computation is needed would go a long way in motivating the importance of the problem. The first paragraph of the paper simply fails to do so, and lacks any references.

[Author Response · NeurIPS 2019]

We'd like to begin by thanking the reviewers for their careful reading of the paper and insightful feedback. It has helped bring to light many typos and some poorly explained sections.

**Reviewer 1**

- **Figure 4:** This figure shows how the $\rho_i$ impact performance. While small $\rho_i$ lead to improved performance, it is critically important what the corresponding $\Delta_i$ are. This is due to the nature of $\tilde{H}_2 = \max_{i \geq 2} {}^i \rho_{(i)}^2 / \Delta_{(i)}^2$: if $\rho_i$ are only small for large $\Delta_i$, there will not be very large gains over $H_2 = \max_{i \geq 2} {}^i / \Delta_{(i)}^2$. If, however, for small $\Delta_i$ we have correspondingly small $\rho_i$ (as is empirically the case on 2 different datasets, as shown in Fig. 4), then large improvements will be realized. We will clarify this figure more in the final version.

- **Theory versus Practice:** We quantify our theoretical gain as $H_2/\tilde{H}_2$ as in lines 84-88. These theoretical gains do not capture the entire picture, only predicting a gain of around 7x over Med-dit for RNA-Seq 20k as opposed to the 50x reduction realized, as our analysis is only able to incorporate pairwise dependence. A lengthier discussion on the other gains we are able to realize and the difficulty in analyzing them can be found in Appendix C.1

- **Error bars on plots:** We will include these in the final version, thank you for the suggestion.

**Reviewer 2**

- **corrSH error rate:** Since corrSH takes a budget as an input, we vary the input budget and plot the error probability for various budgets, noting in the table the smallest budget above which all error probabilities were 0.

- **Typos:** We thank the reviewer for their close reading and pointing out the typos and other unclear portions, these are being corrected for the final version. For example the "middle of the road point" was very poorly characterized; we listed it in the figure caption as $i = 10000$ out of $n = 20000$ (point with the median value of $\{\theta_j\}_{j=1}^n$), but this was a very vague way of referring to it.

- **Remark 3:** Yes, the budget of corrSH is a very important question. Due to the page limit we were forced to relegate many important details to the Appendix; with the extra page allotment for the final version we will make sure to move this remark back to the main text.

- **What yields small $\tilde{H}_2$:** This is a great question that we are currently pursuing; previous works like Med-dit also tried to analyze a similar problem, examining $H = \sum_{i=1}^n \left[ \frac{\log n}{\Delta_i^2} \bigwedge n \right]$. In this work, they took several pages to show that $\mathbb{E}[H] = O(n \log n)$ under the assumption that $\theta_i = N_{(i)}$ where $N_i \sim \mathcal{N}(0,1), i = 1, \ldots, n$ and $N_{(i)}$ denotes the $i$-th order statistic. Note that this is an assumption on the $\theta_i$, and not a true generative assumption on the $\{x_i\}$. Unfortunately these techniques do not translate over, as $H$ is much easier to analyze than $\tilde{H}_2$, as $H$ is a summation over the $\Delta_i$'s rather than a max, and does not involve $\rho_i$'s which are very difficult to analyze.

**Reviewer 3**

- **Problem Motivation:** Algorithms for finding the medoid have gained recent interest in the community; Newling and Fleuret won the best paper award in AI Stats 2017 for their work on a sub-quadratic medoid algorithm [9], and Med-dit followed after this. In addition to the basic medoid, such algorithms are building blocks for $k$-medoid clustering, a commonly used preprocessing step for unlabeled data. Some algorithms for this involve Voronoi iteration, where the medoid of a cluster of points is computed as a subroutine [1]; our scheme could be used to drastically speed up this step. Some alternate algorithms for $k$-medoid clustering are PAM, CLARA, and CLARANS [2]. Our contribution is methodological and goes beyond simply the medoid case, and the methods of correlated sampling we introduced appear to be applicable to these algorithms as well.

- **Lower bounds:** This is a line of ongoing research, as we believe that a lower bound is important and would cleanly close this problem. However, this appears to be highly nontrivial due to the complex dependence structure stemming from the underlying computational problem, as discussed in Sec 2.1 and Appendix C.

# References

[1] H.-S. Park and C.-H. Jun, "A simple and fast algorithm for k-medoids clustering," *Expert systems with applications*, vol. 36, no. 2, pp. 3336–3341, 2009.

[2] L. Kaufman and P. J. Rousseeuw, "Partitioning around medoids (program pam)," *Finding groups in data: an introduction to cluster analysis*, pp. 68–125, 1990.


[Meta-Review · NeurIPS 2019]

The reviewers found the proposed algorithm to be a clever solution to the posed problem and, for the most part, found the document to be well-prepared. The analysis and techniques were not ground-breaking, but the end result was satisfactory